# Oxidative Stress, Inflammation and Colorectal Cancer: An Overview

**DOI:** 10.3390/antiox12040901

**Published:** 2023-04-09

**Authors:** Annamária Bardelčíková, Jindřich Šoltys, Ján Mojžiš

**Affiliations:** 1Department of Pharmacology, Medical Faculty of University of Pavol Jozef Šafárik in Košice, Tr. SNP 1, 040 11 Košice, Slovakia; annamaria.bardelcikova@upjs.sk; 2Institute of Parasitology, Slovak Academy of Science, Hlinkova 3, 040 01 Košice, Slovakia; soltys@saske.sk

**Keywords:** oxidative stress, inflammation, inflammatory bowel disease, colorectal carcinoma

## Abstract

Colorectal cancer (CRC) represents the second leading cause of cancer-related deaths worldwide. The pathogenesis of CRC is a complex multistep process. Among other factors, inflammation and oxidative stress (OS) have been reported to be involved in the initiation and development of CRC. Although OS plays a vital part in the life of all organisms, its long-term effects on the human body may be involved in the development of different chronic diseases, including cancer diseases. Chronic OS can lead to the oxidation of biomolecules (nucleic acids, lipids and proteins) or the activation of inflammatory signaling pathways, resulting in the activation of several transcription factors or the dysregulation of gene and protein expression followed by tumor initiation or cancer cell survival. In addition, it is well known that chronic intestinal diseases such as inflammatory bowel disease (IBD) are associated with an increased risk of cancer, and a link between OS and IBD initiation and progression has been reported. This review focuses on the role of oxidative stress as a causative agent of inflammation in colorectal cancer.

## 1. Introduction

Oxidative stress (OS) is caused by an imbalance between pro-oxidant molecules and the cell’s antioxidant capacity [1]. This imbalance leads to the damage of digestive tract cells, including DNA damage, protein aggregation, and membrane dysfunction [2,3]. It has been proven that reactive oxygen species (ROS) via interaction with cellular macromolecules, including proteins, nucleic acids, and lipids, can disrupt crucial cellular functions. For example, the oxidative damage of DNA may result in bases oxidation, single- and double-strand breaks, or the generation of abasic sites [4]. In addition, unrepaired oxidative DNA damage increases the risk of mutagenesis. These mutations can occur in important genes that regulate cell growth, such as tumor suppressor genes and proto-oncogenes, and can lead to the development of cancer [5,6].

The body’s response to the cell damage of intestinal mucosa exposed to OS is inflammation. Under physiological circumstances, the repair or replacement of inflammation-damaged or dead cells occurs. However, sites repeatedly exposed to inflammation can lead to the development of chronic inflammation and the induction of autoimmune processes [7]. The disruption of homeostasis leads to a breach in the cell integrity and loss of the defensive function in the gut mucosa. Altogether, this leads to mucosal injury and the invasion of pathogenic microorganisms [8]. In some cases, inflammation may precede or accompany tumor development [9]. On the other hand, chronic inflammation can modulate tumorigenesis through the production of reactive oxygen species (ROS). As a result of inflammation, epigenetic alterations occur that promote tumorigenesis through the production of growth factors and pro-inflammatory cytokines [10]. Growing tumors through a feedback pathway further influences the inflammatory environment by the production of cytokines and chemokines.

Oxidative stress is closely associated with inflammatory responses and it has been implicated in the propagation and exacerbation of inflammatory bowel disease (IBD) [11]. IBD is a chronic disease affecting the digestive tract, primarily the large intestine. Several oxidative stress-related IBD genetic risk loci have been identified [2]. There are two primary diseases—ulcerative colitis (UC) and Crohn’s disease (CD) [12], which differ regarding to the extent and involvement in the gastrointestinal tract. Both diseases develop from an overreaction of the immune system. In addition, the conversion of sensitive cells into neoplastically transformed cells in some patients with IBD has been observed [13,14]. Thus, carcinogenesis in the gastrointestinal tract represents a complex pathogenetic process evolving gradually and spontaneously. According to available studies, oxidative stress is significantly involved in the development of colorectal cancer (CRC) [15,16,17,18].

As briefly depicted, the imbalance between ROS generation and detoxification leads to many unfavorable outcomes. This comprehensive review aims to elucidate and discuss the role of oxidative stress in IBD leading to CRC.

## 2. Materials and Methods

The information was collected from articles published from 2010 to 2022 from international and national scientific databases such as PubMed, Medline, Scopus, and Scien-ceDirect using the following keywords “colorectal cancer” and “oxidative stress” or “anti-inflammatory” and “IBD.”

## 3. Oxidative Stress

The human body is constantly confronted with a large number of pro-oxidant molecules. These molecules enter the body from the external environment or are produced as by-products of metabolic reactions that provide essential physiological functions. Pro-oxidant factors include all molecules capable of oxidizing biological substances (lipids, proteins, DNA and carbohydrates). The majority of pro-oxidant molecules are molecules with unpaired electrons, molecules capable of radical production, or molecules facilitating radical production by their catalytic action [2]. Processes catalyzing the radical degradation of molecules, such as ionizing and UV radiation, are also included among the pro-oxidative factors [19]. Of fundamental importance in explaining the role of pro-oxidant molecules in the pathology of human diseases are reactive oxygen species (ROS)—free radicals, nonradical compounds and lipid hydroperoxides. In addition to reactive oxygen-derived species, excessive amounts of reactive nitrogen-derived species (RNS) can be formed in the body during oxidative stress, which can trigger a radical reaction that can damage biologically active molecules [20]. A general overview of the roles of the reactive molecules is provided in Table 1.

ROS are a common part of a living organism’s biochemical processes. The physicochemical properties of pro-oxidant molecules, their importance, and their beneficial or adverse effects have been discussed in many publications. Many cellular and molecular changes mediated by various exogenous and endogenous stimuli are involved in the development of diseases, of which ROS, as the source of oxidative stress, play an important role. ROS are molecules produced by the gradual reduction of oxygen molecules produced by the intracellular organelles, especially in mitochondria [21]. These central organelles are responsible for the generation of a large ROS amount and at the same time are also the most exposed [22]. Thus, mitochondria are the primary source of endogenous ROS. In the process of mitochondrial respiration, molecular oxygen is reduced to water at the complexes involved in the electron transport chain with the concomitant formation of the superoxide radical (O_2_^•−^). In addition to superoxide radical production by the respiratory chain complexes (complexes I, II, III), superoxide radicals are also formed as a result of glycerol 3-phosphate dehydrogenase (mGPDH), 2-oxoglutarate dehydrogenase, pyruvate dehydrogenase and Q oxidoreductase enzymatic activity [23]. Superoxide radicals are generated in the mitochondrial matrix (MM). Manganese superoxide dismutase (MnSOD) found in mitochondria converts the superoxide radical to nonradical hydrogen peroxide (H_2_O_2_) [24]. Hydrogen peroxide reacts in the Fenton reaction with divalent iron by the action of aconitase to form the hydroxyl radical [25]. Complex III and mitochondrial mGPDH in the mitochondrial intermembrane space and cytosol produce a superoxide radical that converts to H_2_O_2_ via superoxide dismutase Cu/Zn SOD [26]. Additionally, H_2_O_2_ is detoxified by catalase (CAT) or the glutathione system, with eight isoforms of gluthationperoxidase (GPx) [27]. GPx1 is a widely available isoform present in the cytoplasm of all mammalian cells [28] that preferentially catalyzes the reduction of H_2_O_2_ to water and oxygen [29]. Moreover, peroxiredoxins (Prxs) also play important roles in the detoxification of H_2_O_2_, alkyl hydroperoxides and peroxynitrite [30].

ROS play an important beneficial role in several physiological processes, including defense against infectious agents and cell signaling when their production is under control [30,31]. Under normal conditions there are enzyme systems involved in the production of ROS in cells. Endogenous ROS may originate in the endoplasmic reticulum, peroxisomes, nucleus or cytosol. This involves some specialized enzymes such as peroxidases, NADPH oxidase (NOX), nitric oxide synthase (NOS), xanthine oxidoreductase (XOR), lipoxygenases (LOXs), cyclooxygenases (COXs) and myeloperoxidase (MPO) [22,32,33].

## 4. Colorectal Carcinoma as a Multilevel Cancer Disease

The inflammatory processes induced by oxidative stress can lead to mitochondrial and neurodegenerative diseases, diabetes, chronic diseases, and aging [34,35]. In addition, it leads to the development of tumorigenesis and tumor angiogenesis processes, which are promoted by free radicals, especially ROS [36]. Increased cancer risk is associated with chronic intestinal diseases [15]. The pathogenesis of colorectal cancer is a complex multistep process that results from long-term inflammation, exposure to infectious agents, or other stressors [37].

IBD is common worldwide, and the number of cases is increasing yearly. While the incidence in children is becoming more frequent, men and women are affected equally [38]. This chronic disease persists throughout life; periods of remission are observed when symptoms are minimal or not existent, and then there are periods of flare-ups when symptoms are most noticeable. Currently, no treatment would completely stop these diseases [39]. Modern medicine aims to achieve the longest possible periods of remission and, in case of flare-ups, suppress the symptoms with symptomatic treatment [40].

### 4.1. Epidemiology

Clinical trial data suggest that the group of patients with IBD has the highest risk of developing CRC compared to the sporadic CRC cases [41,42]. In 2022, 8% of new CRC cases were diagnosed in the US, regardless of gender. The death rate from CRC increased from 2005 to 2019 by 1.2% per year in individuals younger than 50 years and by 0.6% per year in those aged 50 to 54 years [43]. In 2018, Europe had the second-highest incidence of CRC among all diagnosed cases [44]. In 2020, there were 19.3 million new cancer cases worldwide, of which 10.0% were diagnosed as CRC, and the CRC mortality rate was 9.4% [45]. In 2020, only an estimate was established for the population aged 50–80 in Europe, where 13.7% of newly diagnosed CRCs were all from diagnosed cancers [46]. A study (2014) verified a 7% risk of developing colorectal cancer 30 years after the onset of IBD [47]. Statistical studies do not provide information on newly diagnosed cases of colorectal cancer, depending on whether the patient had IBD or not [48]. Patients with IBD receive sufficient attention by monitoring the disease through regular examinations, colonoscopy (every 1–2 years), blood tests and other methods. This group of patients is monitored more often than people without IBD who develop CRC asymptomatically, sporadically, and gradually without knowing it.

### 4.2. Oxidative Stress in Colorectal Cancer Pathogenesis

CRC is a multifactorial disease in which several factors play a significant role. Although the cause of CRC is not yet defined, research results confirm the influence of lifestyle factors, including diet, smoking, stress, alcohol and toxins. Oxidative stress leads to inflammatory reactions of the intestinal mucosa, genetic predisposition, altered intestine immune reaction, and, last but not least, dysbiosis—changes in the composition of the intestinal microbiota [49,50], which are considered an integral part of the CRC development [51].

Many studies confirm the influence of free radicals on the initiation, promotion [52] and formation of IBD [2], and also in the process of multistage carcinogenesis [33]. Oxidative stress in intestinal mucosal cells almost certainly plays a key role in the pathogenesis of CRC. Free radical-induced oxidative damage can result in the activation of metabolic pathways, during which other proteins affecting the processes of cell proliferation and inflammation are created.

The effects of oxidative stress on the cells of the colon mucosa could be divided into three levels: (a) the level of biological membranes (oxidation of lipids), (b) the level of the nucleus (oxidative DNA damage), and (c) the level of proteins and carbohydrates. At the same time, products of oxidative damage by free radicals represent potential indicators or markers of CRC outcome [15].

Lipid peroxidation, the main feature of oxidative stress, promotes cell destruction at the level of phospholipid cell membranes. The endoplasmic reticulum is a reservoir of calcium ions, which escape into the cytoplasm due to the peroxidation of membrane lipids. As a result, there is a loss of control over the activity of Ca^2+^-dependent enzymes, whose activity is controlled by the levels of calcium ions in the cytoplasm [53]. Moreover, increased levels of Ca^2+^ ions in the cytoplasm stimulate the NO synthase (NOS) to produce the NO, which induces oxidative damage [54]. Mitochondrial lipids are extremely important for maintaining structural integrity and mitochondrial functions [55], where oxidative damage to mitochondrial membranes disrupts cell energy metabolism [56]. Damage to the phospholipid bilayer of the cytoplasmic membrane of colon cells leads to malfunctions of membrane receptors, the release of small molecules into the extracellular environment, and subsequent membrane rupture. As a result of lipid peroxidation by free radicals, the structure of fatty acids is damaged and their function is lost. In addition to the formation of by-products such as gaseous alkanes—ethane, propane, pentane, and hexane, lipid peroxidation produces highly toxic aldehydes, ketones, hydroxy aldehydes and epoxides. Elevated levels of ethane, methane and pentane have been detected in patients with Crohn’s disease and ulcerative colitis [57].

The final product of lipoperoxidation is malondialdehyde (MDA), which reacts with DNA to form MDA–DNA complexes. MDA–DNA complexes have been shown to have pro-mutagenic properties and induce mutations in oncogene/tumor suppressor genes in human tumors [58,59].

Enzymes such as lipooxygenase and cyclooxygenase are involved in oxidative stress as well. Lipooxygenase ensures the synthesis of hydroperoxides, while cyclooxygenase ensures the synthesis of endoperoxides, from which prostaglandins are formed [60]. Cholesterol derivatives have significant pro-inflammatory and pro-apoptotic effects. Free radicals oxidize cholesterol to form oxysterols (7α-OH or 7β-OH), which are further oxidized to 7-keto-cholesterol and toxic C-5 and C-6 oxygenated derivatives of cholesterol [61].

As oxidation products of lipids and carbohydrates, ROS, RNS, and metal ions participate in protein oxidation. Proteins with a side chain composed of amino acids containing sulfur atoms (methionine, cysteine) are easily oxidizable. While the oxidation of cysteine produces disulfides, the oxidation of methionine produces methionine sulfoxide. Hydroxyl radicals activate the peptide bond, forming carbon radicals that react with oxygen [62], which creates an alkyl peroxyl radical, an alkyl peroxide or an alkyl radical. These radicals also oxidize other places on the polypeptide chain [63]. In the absence of oxygen, carbon radicals of two different proteins due to the formation of cross-links are responsible for breaking the secondary structure of proteins, which gives rise to protein aggregates resistant to degradation by proteolytic enzymes [64]. Significant markers of oxidative damage are carbonylated proteins with incorporated carbonyl groups. Increased levels of carbonylated proteins have been detected in patients with ulcerative colitis [65], diabetes mellitus [66], and Alzheimer’s disease [67]. Protein oxidation leads to a gradual loss of the structure and biological function of enzymes, receptors and structural proteins [68].

Free radicals (ROS/RNS), ionizing radiation, and transition metals may directly damage the DNA/RNA. Oxidative DNA damage results in DNA strand breaks, DNA fragmentation, and base mismatches, leading to unwanted mutations [69]. These are subsequently repaired by a system of repair enzymes that cut out and simultaneously replace the damaged bases with new bases. Non-specific endonucleases remove the entire chain. Specific DNA glycosylases remove one specific damaged base. The oxidation of DNA also changes the primary structure of DNA, the exchange or loss of bases and the formation of cross-links [70].

### 4.3. Mechanisms of CRC Development Induced by Oxidative Stress

Preclinical and clinical research has identified the primary mechanisms by which free radicals contribute to the development of CRC. The development of CRC is a multistep process of transforming a healthy intestinal cell into an abnormal one, where one mutation is not enough to cause the CRC. The direct oxidizing of bases, sugar components, and proteins associated with DNA causes mutations where free radicals, via transcription factors Nrf2 and NF-κB, intervene with inflammation and carcinogenesis [71,72,73]. The activation/inhibition of nuclear factor erythroid 2-related factor 2 (Nrf2), activated by free radicals, is considered effective in CRC prevention and treatment [71]. Its activation inhibits oxidative stress and inflammation, resulting in the prevention CRC development [71]. The primary function of Nfr2 is the regulation of cytoprotective and antioxidant gene expression. Under normal conditions, Nrf2 is in a complex with the inhibitory proteins Keap1 via the ETBE and DLG domains. Keap1 proteins enable the ubiquitination of the Nrf2 protein and, subsequently, its degradation in the proteasome. Keap1 proteins represent a regulatory mechanism by which the amount of Nrf2 in the cell’s cytoplasm is regulated. Protein modifications play a key role in adaptation to oxidative stress by activating antioxidant or metabolic programs to counteract ROS metabolism [74,75].

The promoter hypermethylation of Keap1 leads to a reduction of Keap1 expression and Nrf2 accumulation in the nucleus of patients with CRC [76]. As a result of oxidative stress, Keap1 proteins dissociate from Nrf2 and enter the nucleus. Nrf2, together with the small sMAF (small musculoaponeurotic fibrosarcoma oncogene homolog) protein, induces the transcription of antioxidant response elements (ARE) [77] and leads to the expression of more than 500 target genes, including antioxidant enzymes [78,79] such as NAD(P)H: quinone oxidoreductase-1 (NQO1), heme oxygenase (HO-1), superoxide dismutase 1 (SOD), and CAT; and enzymes involved in glutathione metabolisms such as glutathione S-transferase (GST), GPX and others [80,81]. Nfr2 eliminates ROS through the upregulation of enzymes involved in the induction and synthesis of antioxidant molecules [80]. Heme oxygenase 1 (HO-1) catalyzes the degradation of heme to iron, biliverdin and carbon monoxide (CO) [62]. CO suppresses the nuclear translocation of NF-κB p65, which plays a central role in the inflammation process [82,83]. Its activation leads to the production of pro-inflammatory cytokines (TNFα, IL-1β, IL-6), chemokines (MCP-1, MIP-1, RANTES, eoxantin, IL-8), transcription factors (Jnk, Erk, p38), inflammation mediators (COX-2), antimicrobial peptides and adhesive molecules (ICAM-1, VCAM-1, ELAM) [84]. Therefore, by inhibiting the nuclear translocation of NF-κB, there is a decrease in the intracellular level of pro-inflammatory cytokines.

On the other hand, overexpression of Nrf2 can promote colorectal tumor growth. The aberrant activation or accumulation of Nrf2 is connected with malignant progression, chemotherapy resistance, and poor prognosis [85,86]. Therefore, if the tumor has already occurred, Nrf2 inhibitors are administered as anticancer agents. Effective Nrf2 inhibitors are brusatol [87], chrysin [88], trigonelline [89], ascorbic acid [90] and retinoic acid [91]. Luteolin, as an inhibitor of Nrf2, reverses the sensitivity of colorectal cancer cells to chemotherapeutic agents [92].

Nrf2 is probably also an important inhibitor of metalloproteinases (MMPs). While in humans, Nrf2 activation inhibits MMP-7, and in Nrf2-deficient mice, the level of MMP-3 is higher than in controls [93]. At the same time, the Nrf2-deficient mice are more susceptible to benzo[α]pyrene-induced tumor formation [94]. The pathogenesis of CRC is closely related to oxidative DNA damage and the production of pro-inflammatory cytokines, overexpression of Nrf2, expression of metastasis-associated colon cancer 1 (MACC1), and stimulation of MMP production via TNFα [95]. Long-term stimulation of the intestinal epithelium by inflammatory cytokines and persistent activation of NF-κB are involved in the development of chronic inflammation and the initiation of carcinogenesis. The immune system responds to signaled inflammation by activating T-cells and infiltration of inflammatory neutrophils into the mucosal layer of the intestine. Neutrophils produce large amounts of ROS/RNS, whose high local concentration damages other cells of the intestinal mucosa [96]. TNFα together with IL-1β stimulates matrix metalloproteinase (MMP) production and simultaneously regulates the COX-2 overexpression in the early stages of carcinogenesis [97]. IL-6 activates the JAK/STAT pathways, leads to the inhibition of apoptosis and, together with TNFα, promotes angiogenesis and tumor growth [98]. Oxidative DNA damage represents the beginning of the transformation of intestinal epithelial cells. The subsequent activation of oncogenic genes provides cells with advantages in the form of unregulated proliferation, growth, resistance to apoptosis and survival [99].

The inflammatory environment contributes to tumor initiation by producing reactive oxygen/nitrogen species or epigenetic changes (e.g., DNA methylation, histone modifications or changes in chromatin organization) that can play a role in carcinogenesis by silencing the expression of tumor suppressor genes and activating oncogenic signaling [100,101]. It also promotes tumorigenesis by providing growth factors and pro-inflammatory cytokines [10]. The environment of chronic inflammation as a result of the ROS signaling function provides transformed cells with suitable conditions (energy source and metabolites) to initiate carcinogenesis. The resulting effect of ROS on tumor initiation and promotion is related to quantity, location and duration [74].

Under normal conditions, ROS regulate many signal transduction pathways. In general, tumor cells have a higher level of ROS than healthy cells. On the other hand, cancer cells tend to produce higher levels of antioxidants to counteract the damaging effects of ROS. This suggests that maintaining a certain level of ROS is essential for cancer cells to function properly [102]. Thus, while low levels of ROS can promote cell proliferation and invasion, excessive levels of ROS cause oxidative damage to proteins, lipids, RNA and DNA, which in turn induces cell death [103,104,105]. As a result of metabolic abnormalities and oncogenic signaling, the protective mechanism against the persistent oxidative stress of the tumor cell is activated. This redox adaptation reaction of cancer cells results in drug resistance [15].

Screening of oxidative stress markers and antioxidants in colorectal cancer patients suggests the existence of a protective mechanism for the tumor cell. The study of Burwaiss et al. [106] analyzed ROS in tumor cells in adjacent surrounding tumor tissues from patients with colorectal cancer and adjacent normal tissues. They found that the tumor’s oxidant and antioxidant levels were significantly lower than those in the surrounding tumor tissue and control healthy tissue. In addition, Indran and co-workers [107] reported reduced both basal and H_2_O_2_-induced ROS production in HeLa cells with overexpressed human telomerase reverse transcriptase (hTERT), indicating a possible link between hTERT and OS in cancer cells. hTERT is a significant characteristic of CRC and has a crucial role in the maintenance and the synthesis of chromosomal ends—telomeres [108,109]. Telomerase has non-telomeric function and supports growth factor-independent growth [110,111]. Elevated telomerase activity is reported in almost all human cancers [111]. The transcription factor YBX1 (cancer-related gene) upregulates the activation of the Nrf2 gene promoter in the presence of hTERT, which reduces ROS in CRC cells, thus promoting cancer progression [112] (Figure 1). Increased telomerase activity in cancer has been shown to promote resistance to apoptosis. In addition to the response to oxidative stress, cell growth and proliferation, hTERT also regulates a wide range of important cellular functions, such as gene expression, signal transduction, and mitochondrial function [113]. These effects are called non-canonical functions of hTERT [114]. In addition to its nuclear localization, hTERT is also found in cytoplasm and mitochondria [115]. hTERT transport into and out of organelles is regulated by a nuclear targeting signal sequence and a mitochondrial targeting sequence [116]. The secretion of hTERT from the nucleus into the mitochondrion is induced by oxidative stress [117]. Ahmed et al. [118] reported in their study that hTERT overexpression improved mitochondrial functions by inhibiting ROS production and increasing mitochondrial membrane potential in MRC-5 lung fibroblast mitochondria. In cancer cells, hTERT translocation and overexpression improves mitochondrial potential, enhances respiratory chain activity, protects mitochondria from environmental damage, and decreases reactive oxygen species production, ultimately leading to survival [119,120,121].

In addition to controlling the cellular processes of free radical regulation by inhibiting oxidative stress and inflammation, new Nrf2 target genes have been identified as being involved in the inhibition of cell proliferation and the induction of apoptosis [73].

The activation of antioxidant enzymes is the primary cell defense mechanism. The overall increase in SOD activity is a response to tissue protection against oxidative damage under conditions of inflammation and oxidative stress in the pathogenesis of IBD. Accordingly, SOD levels in the peripheral blood of IBD patients are already being used as bio-markers of oxidative stress [122]. Both SOD1 and SOD2 protect against spontaneous tumorigenesis, and although they have been described as tumor suppressors, they can also be upregulated during tumorigenesis [123]. In several models, SOD and GPX, by reducing the hydrogen peroxide to water, can protect against tumor initiation induced by carcinogens and ROS [124,125]. Malinowska et al. [126] have studied the activities of GPX and SOD in the erythrocytes of CRC patients. The results showed a statistically increased activity of SOD and GPX. In mouse models of colon cancer, GPX3 has been found to suppress tumor initiation [127]. Meanwhile, mice with a reduced expression of SOD2, either alone or in combination with a loss of GPX1, showed increased DNA damage and tumor incidence [128,129]. However, the tumor initiation effect of GPX was also confirmed. GPX2-deficient mice were protected from azoxymethane-induced colorectal tumorigenesis, which is demonstrated by the tumor-initiating activity of the antioxidant GPX2 [130]. Furthermore, other authors reported that GPX2 overexpression also plays a role in the development of prostate cancer [131].

The study of Kundatepe et al. [132] monitored the oxidative stress parameters protein carbonyl (PCO), advanced protein oxidation products (AOPPs), malondialdehyde (MDA), total nitric oxide (NOx), pro-oxidant-antioxidant balance (PAB), and the ferric reducing of antioxidant power (FRAP), and found that in an impaired oxidative/antioxidant condition in breast cancer (BC) and colon cancer (CC) the oxidative stress is favored. A German study lasting from 2003 to 2012 evaluated two biomarkers of oxidative stress in CRC patient (n = 3361) d-ROMs (Diacron’s reactive oxygen metabolites) and TTLs (total thiol levels) [16]. A strong association between higher d-ROMs and lower TTL levels was observed with poorer survival. The ratio of TTL to d-ROM was an even stronger predictor of CRC prognosis than TTL alone. The results in this study showed a significant improvement in the prediction of CRC prognosis for all cancer stages. The study suggests that oxidative stress contributes significantly to premature mortality in CRC patients. Work by Sawai et al. [133] focused on the correlation of d-ROM and the neutrophil-to-lymphocyte ratio (NLR), an inflammatory marker, as possible prognostic markers. The results obtained during the period 2013–2018 indicate that CRC patients (n = 163) with high d-ROM and high NLR had the worst disease-specific survival. Simultaneously, they discovered that tumor size was significantly associated with d-ROM and NLR. The combination of d-ROMs and NLR as prognostic markers in colorectal cancer may effectively predict prognosis in CRC patients.

Many scientific studies state that the failure of antioxidant mechanisms leads to cancer initiation and subsequent promotion, including CRC [134]. Total antioxidant activities in CRC patients suggest this. Zinczuk et al. [135] showed a low activity of CAT, the enzymes responsible for the elimination of hydrogen peroxide, in the blood of patients with CRC. At the same time, they detected a high activity of superoxide dismutase SOD and a higher concentration of uric acid as the most important plasma non-enzymatic antioxidant in patients with colorectal cancer compared to healthy patients. Simultaneously, the markers of oxidative stress such as MDA, advanced glycation end products and advanced oxidation protein products are significantly increased in CRC patients. On the basis of these results, it is suggested that oxidation processes exceed the antioxidant defense.

Cell transformation is associated with the transcription inhibition of apoptosis-related genes, such as cellular inhibitors of apoptosis (cIAPs), caspase-8/FADD-like IL-1beta-converting enzyme inhibitory protein (c-FLIP) and members of the bcl2 family (e.g., A1/BFL1 and bcl-xl) [136].

### 4.4. Microbial Dysbiosis and Colorectal Carcinoma

Many microorganisms live in the intestinal lumen and on the intestinal mucosa [137]. In addition to a health-promoting effect, certain members of gut microbiota can be a source of oxidants and contribute to the development of CRC [138]. These contribute to mucosal inflammation, which leads to significant changes in the bacterial population of the large intestine [139]. Changes associated with a reduction in the number and a change in the overall diversity of bacterial species could contribute to inappropriate reactions of the intestinal immune system and thus be key in the development of IBD and carcinogenesis [140,141].

Some potential bacteria associated with sporadic CRC are *Streptococcus bovis*, *Streptococcus gallolyticus*, enterotoxigenic *Bacteroides fragilis*, *Fusobacterium nucleatum*, *Enterococcus feacalis*, *Escherichia coli*, *Peptostreptococcus anaerobius* and *Salmonella* sp. [142,143]. They have been associated with CRC and cause damage to the host DNA by genotoxic agents, including a colibactin secreted by *Escherichia coli*, a toxin produced by *Bacteroides fragilis* and a stomach toxin from *Salmonella* [139]. *E. coli* are believed to play a primary role in the induction of chronic inflammation [144]. Their lipopolysaccharides are known to increase the expression of Toll-like receptor 4 (TLR4), which leads to the initiation of CRC [145]. Subsequently, there is an overexpression of NF-κB, which contributes to inflammation and the development of CRC. *Streptococcus bovis* as an early sign of CRC [146], and *Fusobacterium nucleatum* as an indicator of a worse prognosis in CRC patients, increase the inflammation level, leading to the development of CRC [147]. Both in vitro and in vivo studies confirm a high risk of CRC in connection with *Enterococcus faecalis*, which stimulates macrophages to produce superoxides, and together with hydroxyl radicals, significantly damages DNA [148]. *Peptostreptococcus anaerobius* on intestinal epithelial cells activates TLR2/TLR4 and increases intracellular ROS levels, which promotes cholesterol synthesis and cell proliferation [142,149]. The relation of tumorigenic bacteria to the development of CRC is thoroughly explained by a review from Li et al. [139].

Reactive oxygen species represent a risk factor for the development of CRC, as they cause oxidative damage to the intestinal mucosa cells, leading to the breakdown of the intestinal barrier and its dysfunction. Oxidative damage leads to abnormal Paneth cell morphology, the dilatation of the crypt lumen and erosion of its villi, and an increase in the apoptosis of crypt cells and homogeneously electron-lucent granules [150,151], which creates an entry gate for pathogenic microorganisms. The disruption of the barrier triggers the invasion of commensal and pathogenic microorganisms and leads to contact between intestinal epithelial cells and components of the microbiota, which may have protumorigenic properties [10].

### 4.5. Markers of Oxidative Stress

Teams of scientists are trying to find suitable markers for indicating the degree of damage to the colon cell by oxidative stress and the stage of the disease that the patient is in. In humans, different methods of oxidative stress measurement are used in materials such as tissue, urine, blood or serum. Regarding precancerous conditions, it is essential to detect the level of initial damage to the cells or macromolecules by oxidative stress.

Reactive species may react with DNA, lipids and proteins. Molecules such as 8-oxoguanine, 8-hydroguanine, 8-hydroxy-deoxy-guanosine and others are leading indicators of oxidative DNA damage. For oxidative damage in protein samples, it is necessary to obtain tissue to detect markers such as 2-hexahistidine, carbonyl groups, hydroperoxides, protein-bound DOPA, 3-nitrotyrosine, etc. [152].

Nevertheless, intensive research on biomarkers as primary indicators of oxidative damage to the body with CRC is underway. CRC-specific markers could also determine the grade/stage of cancer. In 1968, a tumor marker—a carcinoembryonic antigen (CEA)—was discovered while isolating extracts from the liver metastasized by colorectal cancer and normal fetal digestive tract [153]. CEA is associated with many types of cancer, predominantly with gastrointestinal tumors. The study of Chandramathi et al. [154] detected a significantly increased level of the advanced oxidative protein product (AOPP), hydrogen peroxide (H_2_O_2_) and MDA in the urine of CRC patients. The determination of CRC markers was also addressed in a study from 2019, in which the level of MDA in the blood was significantly higher in CRC patients compared to healthy controls. The authors stated that blood catalase (CAT) and MDA could be used in CRC diagnostics or as indicators of tumor invasion depth and the presence of lymph node metastasis [135]. Depending on the stage of the disease characterized by the relevant biomarker, the survival of patients diagnosed with CRC could be predicted.

In addition to oxidative damage products, the factors of the antioxidant system are also determined. As part of monitoring the state of antioxidant capabilities, the activities of antioxidant enzymes are also detected. The most important cellular protective mechanisms against ROS are antioxidant enzymes such as catalase, glutathione peroxidase, and Prxs [155,156]. In another study, the activities of Cu/Zn-SOD, GPx, GR and GSH activities, and the concentration of non-enzymatic antioxidant uric acid (UA), were detected in the serum or plasma in patients with CRC. Results of this study showed a significant increase in both enzymes activity and UA concentration. In contrast, CAT activity was considerably lower in the serum of patients with colorectal cancer compared to the control group [135]. Markers such as CAT, AOPP, H_2_O_2_ and MDA could represent non-invasive oxidative stress markers in colorectal cancer.

However, the determination of antioxidant enzyme activities is limited because not all antioxidant enzymes and their mutual antioxidant action are known yet. Due to this fact, the preferred methods for determining the total antioxidant capacity of the investigated material are total antioxidant capacity (TAC), total oxidant status (TOS), oxidative stress index (OSI) and non-enzymatic ferric reducing ability of plasma (FRAP). The results of clinical studies clearly demonstrated significantly lower levels of TAC and FRAP in CRC patients’ plasma and a significant increase in TOS and OSI levels compared to the control group [135].

## 5. Conclusions

In general, the cause of cancer is long-term exposure to various carcinogenic factors, including inflammation. The results of many clinical studies are controversial regarding oxidative stress research in relation to the detection and therapy of colorectal cancer. Several preclinical and clinical studies show that different endogenous and exogenous antioxidant molecules effectively reduce oxidative stress in the intestine. However, no study confirms a cure with the use of antioxidant therapy.

As a result of proving the association between inflammation and the development of cancer, research is devoted to the development of anti-inflammatory drugs, which play an important role in the treatment and prevention of cancer. It is evident that substances with an antioxidant effect can also have pro-oxidant effects under certain circumstances.

Many disputed issues persist and are far from being definitively resolved when assessing oxidative stress in CRC carcinogenesis as well as the role of antioxidants in antitumor protection.

This review includes the results of important publications, including new knowledge in connection with carcinogenesis induced by oxidative stress in IBD patients and spontaneously induced by carcinogenesis.

## Figures and Tables

**Figure 1 antioxidants-12-00901-f001:**
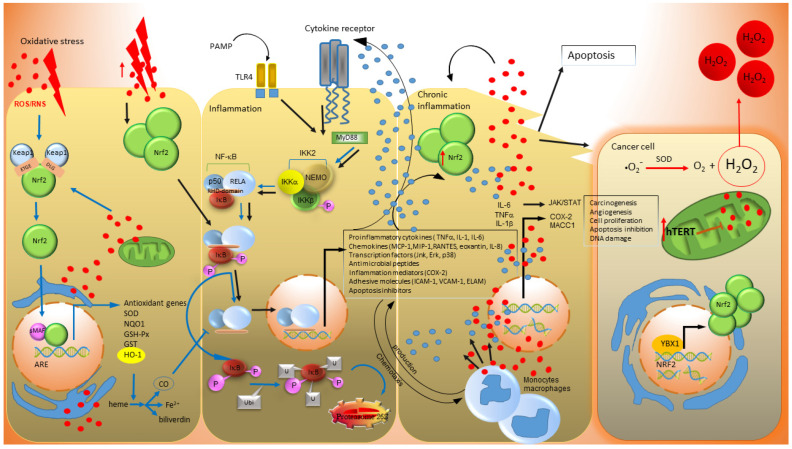
Mechanism of colorectal carcinogenesis. In the phase when the colon cell is exposed to oxidative stress (OS), the Nrf2 protein is activated. Its level in the cell’s cytoplasm increases, which correlates with an increase in the cell’s antioxidant capacity and content of antioxidant enzymes (e.g., HO-1). The antioxidant products prevent NF-κB from entering the nucleus by inhibiting pro-inflammatory cytokines. This is the one of several mechanisms by which the cell protects itself from OS. However, if the cell goes into the inflammation phase, the level of Nrf2 increases to such an extent that there is a nuclear translocation of NF-κB, leading to the massive production of pro-inflammatory cytokines. The result is inflammation, since if the cell of the intestinal mucosa can no longer defend itself and exhausts its antioxidant capacity, the cell enters a phase of chronic inflammation. Moreover, oxidative modifications of DNA may result in alterations in bases, strand breaks, the inactivation of tumor suppressor genes, or the overexpression of proto-oncogenes. The damaged cell subsequently enters apoptosis or transforms into a tumor cell. In a transformed cell, the auto- or self-antioxidant system hTERT is activated, reducing the ROS. This mechanism could be considered a protective mechanism by the tumor cell to ensure its existence and proliferation. ROS produced by the tumor cell are subsequently transported out of the cell and thus cause oxidative stress to surrounding cells. At the same time, extracellular ROS also influence the microenvironment to ensure the production of energy and metabolites for tumor cells. Blue arrows show the regulatory mechanism of cell protection against OS. Black arrows show the effect of excessive OS on a cell during the transformation into a tumor cell. The red dots represent ROS/RNS, while the blue dots depict inflammatory cytokines and chemokines. Pathogen-associated molecular patterns (PAMP molecules): lipopolysaccharides, toxins, lipoteichoic acid, peptidoglycans, lipoproteins, flagellin, glucanases, chitin and viruses. The original figure for this review was made by the Zoner Photo Studio X (Zoner a.s., Slovakia).

**Table 1 antioxidants-12-00901-t001:** Overview of reactive molecules causing oxidative stress.

Reactive oxygen species (ROS)	free radicals	superoxide (O_2_^•−^) hydroxyl radicals (HO^•^) peroxyl (RO_2_) alkoxyl (RO) hydroperoxyl (HO_2_)
	lipid hydroperoxides	
	nonradical	singlet oxygen (^1^O_2_) hydrogen peroxide (H_2_O_2_) hypochlorous acid (HClO) chloramines (RNHCl) ozone (O_3_)
Reactive nitrogen species (RNS)	radical	nitric oxide (NO), nitrogen dioxide (NO_2_)
	nonradical	peroxynitrite (ONOO¯) dinitrogen trioxide (N_2_O_3_)

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
