# Peer review of "Oxidative Stress, Inflammation and Colorectal Cancer: An Overview"

_antioxidants, 2023, doi:10.3390/antiox12040901_

Round 1
Reviewer 1 Report
The review is interesting, well written and structured and informative with the majority of statements being complete and correct. However, there are a number of changes in content and language require to improve clarity and factual correctness. I list them below.
1. Line 25: There is no accumulation of ox. stress possible since ROS have an extremely short lifespan. Just the caused damage is able to accumulate. Please correct.
2. When you talk about inflammation being the result in intestinal mucosa, don't you rather ignore any physical damage to various cellular components such as for example DNA damage (breaks, oxidative modification of nitrogen bases such as 8-oxdG) which can result in known mutations (APC, p53 etc) which are involved in the initiation and progression of CRC. Please add this to your statement, even if your focus is on inflammation in your review and you correctly describe this further down in your review (lines 196/7). You should still get the balance right.
3. Lines 27 and 35: You mention both cases: ox stress causing inflammation and inflammation resulting in ox. stress. You should emphasise this complex relationship better.
4. line 46: I don't think it is correct to say that "patients induce...processes" as this formulation sounds as if the patients do this consciously. Please rephrase to state that this happens "IN patients" by molecular mechanisms.
5. the statement in line 50 requires appropriate references evidencing such studies unless you refer to other places in the review here for more detailed information.
6. line 51: The term "imbalance in oxidative stress" is not correct as ox stress is the result of an imbalance between generation and scavenging/detoxification of ROS by antioxidant systems. Please correct and rephrase.
7. line 92: as mentioned above? Radicals are UNABLE to accumulate due to their extremely short lifespan. Also, only from some mitochondrial sites superoxide is moved towards the matrix (except complex III), while the other site directs radicals towards the membrane. Please correct.
8. line 97: It should be Cu/Zn SOD which is SOD2 and uses both molecules. I am not aware that these are separate molecules. It might be strangely written in reference with a comma between Cu and Zn.
9. The statements in lines 111-117 require references, also line 130.
10. line 173: You probably mean "oncogenes/tumour SUPPRESSORS". Please correct.
11. line 188: please name these 2 proteins, otherwise it is not informative.
12. lines 201: Please add the modification of DNA bases, for example 8-oxodG as a very common one.
13. line 209: carcinogenesis is a process and has NO enzymatic activity. Please rephrase.
14. lines 212/3: How does the antioxidant capability of NRF2 inhibit proliferation? Importantly, just a stop in proliferation does not result in apoptosis but quiescence. Please better describe the mechanism how apoptosis is induced and provide an appropriate reference for it. You correctly describe in the figure legend line 262 that it is the damage that results in apoptosis which makes much more sense.
15. line 220: In my view the term "facilitate" is not correct here as it rather means "promoting" while rather the opposite (inhibition of ox stress) is the case. Please rephrase.
16. line 231: I think that your NFkappaB protein p6 should rather be p65. Please correct.
17. line 235: please add "nuclear" to "translocation"
18. line 237: I don't think you have described any link between NRF2 and NFkappaB, thus you cannot draw such a conclusion without any proper evidence. Please add as for example explained in figure legend 1 lines 257/8.
19. line 243: "which damages DNA" does not bear any reference to the rest of the sentence where you described the influence or radicals onto cells. Please rephrase and correct grammar to generate a complete and correct sentence.
20. line 250: resistance to WHAT?
21. line 254: You have to give evidence which "information obtained" you refer to-either by citing references or referring to the previous text.
22. line 258: If you claim "this is the mechanism how the cell protects itself" you exclude any other possible mechanisms/pathways which certainly exist as there are many other transcription factors that are able to induce antioxidant factors.
23. line 263: You describe "hTERT" as an antioxidant system without firstly explaining that hTERT is the protein part of human telomerase and without giving any references to this non-canonical function of hTERT within mitochondria which is described only later in the text. Since this function of TERT is non-telomeric it is not predominantly associated with cell proliferation as is the canonical, telomere-maintaining function. Moreover, there are also tumours without telomerase (activity) which either activate an alternative telomere lengthening programme (ALT mechanism) or have sufficiently long telomeres, like for example various childhood cancers, for example retinoblastomas and neuroblastomas.
24. line 266: Importantly, extracellular ROS are also used by tumour cells to signal to the microenvironment, for example to cancer-associated fibroblasts to produce energy and metabolites for the cancer cells. Please add this important signalling function of ROS for cancer cell/microenvironment communication. Also add to line 281.
25. line 277 and above: please better describe how inflammation modulates epigenetics.
26. lines 285/6: What exactly do you mean under "redox adaptation" and how does this cause drug resistance? There are many different pathways how cells perform drug resistance. Please correct and rephrase.
27. line 286: In addition to high hTERT expression there is also a constitutively high level of telomerase activity (TA) which in its canonical function maintains telomeres. Please add this important fact.
28. Importantly, the interaction of hTERT with NRF2 is by no means THE mechanism how hTERT contributes to tumourigenesis. The first and foremost is that high TA maintains telomere length as an important prerequisite of ongoing proliferation capability of tumour cells. Secondly. hTERT is able to localise to mitochondria under stress where it decreases oxidative stress, drug sensitivity and apoptosis level. As a reference you can cite for example Ahmed et al., 2008.
29. Line 288: I am not sure that cancer cells have lower ROS than normal cells. It could be the opposite. Please check and bear in mind that ref. 75 is just one of many pathways and mechanisms of promoting tumour progression. There are many more and you cite one (ref. 95) with a completely different result! Thus, it is important not to overinterpret single studies, but to describe the big picture and then you can in addition discuss deviations and contradictive findings.
30. Likewise, I am not convinced that the study in reference 76 is representative, moreover that it was published in a Chemistry journal. Please check more studies.
31. line 315: the statement requires a reference as if ref. 90 measured the activity in erythrocytes, this does not say anything about the levels in CRCs. Also, the statement in line 318 needs to state in what cell type this result was found (probably erythrocytes from CRC patients", this is important to avoid any misleading information.
32. What study analysed GPX in line 319? Was it also ref. 91? Please state as otherwise it could have been any study which needs to be referenced.
33. line 329: please describe what type of patients was analysed in the study.
34. line 346: please provide the reference for Zinczuk et al here [99] which you do not need below as the next sentence relates to this clearly.
35. line 350: rather than a "weakened antioxidant barrier" it seems that the balance between ROS generation and antioxidant defence was moved towards the former. There is no active influence of ROS levels on antioxidants! Please correct.
36. line 353: You now claim that ALL ox stress is from gut microbiota, but you never mentioned this above and in my view is not correct. It could be one of several because mitochondria or NOX from intestinal cells could be another. In that case it should be "A source" instead of "THE source" which implicates the ONLY source. Please clarify in any case from which sources ROS are produced here and emphasise this already in the beginning of the review.
37. line 354: inflammation of what? Mucosal cells? Systemic etc?
38. lines 351-58: please provide references.
39. line 389: What is a "method of ox stress"? Do you mean of measurenet of ox stress? Please correct.
40.
40. line 390: What about gut biopsies?
41. lines 391400: "damage to the organism/body" is a bit general since the damage is rather to cells or macromolecules. Also "on time" is not really clear. Do you mean In time to be able to intervene with tumour initiation/progression? Please specify.
42. line 394: give reference after Dalal et al. (111?).
43. lines 418/20: please specify which SOD you talk about and perhaps say "dismutaseS" if you mean all of them (there are 3).
44. line 421: Higher in what/where? Please specify.
45. lines 423-6: The grammar of this sentence is wrong and distorts the content since NOT the GPXs develop the diseases, but the KO mice. Please rephrase and correct content.
language mistakes: Remove "the" in lines 24 and 29. line 56 "such AS...". line 63 :it should be "SUBSTANCE oxidation" no plural required for the 1st noun. line 66: better: "Prooxidant molecules". line 68: "OF fundamental importance..." line 69: remove "the" in front of ROS. line 74: better" roleS of reactive molecules" since there is more than 1 role. line 80: remove first "disease" since it is redundant. line 80: Please remove "the" and "cell" as it is already contains in "intracellular". line 91: please place ref, 17 to the end of the sentence and not in the middle of some words. line 103 "as A ...disease". line 114: "..or not EXISTENT", otherwise grammar is wrong. line 114: remove "the". line 146 "radical-induced" no plural required. line 161: "and MITOCHONDRIAL functions", no "the" required. line 203: I would suggest to say "mechanismS" as there is more than one. However, "mechanisms of CRC" is not a good term as CRCs are not mechanisms. What you rather mean is to say is by what mechanisms ROS promote CRCs. Please rephrase the heading. line 208 "oxidizing.... CAUSES".. lines 211& 214: remove "the". line 222: better "OF patients". lines 244/5 "Tnfalpha ....STIMULATES...and REGULATRS..." line 255: Have you introduced the abbreviation OS before? Please do here or before. line 259: what is NRF2p? Please explain. If you mean p=protein, better spell it out. line 265: remove "The" in front of ROS. line 266: please remove "the" in front of "surrounding. line 269: Please explain the abbreviation PAMP. line 270: I think there should be a comma between "lipoproteins" and "flagellin" as otherwise it reads like a list of lipoprotein and a virus is certainly none and glucanases neither. Moreover, it should rather be "VIRUSES" in plural and the last comma should be replaced by an "and". line 274: Please doe not use a capital "C" for "cellular" and "caspase". line 282: "ROS REGULATE" as "species"is plural. line 295: please remove "the" in front of "NRF2 inhibitors". line 297: Replace comma after [82] with "and". line 298: better" chemoTHERAPEUTIC agents". line 310: please divide as BIO-MARKERS. line 313: Please remove "the" in front of SOD and replace the comma after SOD with "and". line 328-330: grammar of this sentence is wrong as the st and the 2nd part do not fit. Better divide with a dot after "TTL" and add reference 96 here already and not just after several sentences. line 332: "a strong ASSOCIATION" "a"=singular! line 333"was AN EVEN stronger..." line 335: "...suggestS". line 338 "during THE period...". lines 343 and 393: "works" is not used in this context in English, please replace with "studies". line 344: If CRC is included in the study (otherwise you should not state it here), the reference 98 should be at the end of the sentence. line 371: In vivo and in vitro should be as latin words be in italic. line 376: better to say "..by a review from Li et al.,...line 387: Better say "TEAMS of SCIENTISTES" with no article. line 389: "in which the patient is IN". line 393: grammar/language in incorrect: better say: "oxidative stress-induced damage to... macromolecules"... lines 3945: Remove "The" in front of "molecules" and "leading". line 420: "of THE antioxidant uric acid.." . line 423: "peroxidase 1" as there is just this one. line 425: was the abbreviation for AOPP explained already? line 427 you could also say "enzyme activities". line 445: better" proOXIDANT". 446: better than "points" use "issues/topics" . line 449: Instead of "The worj" better say "THIS REVIEW".
Reviewer 2 Report
In this review, the authors summarize information published from 2010 to 2022 under the keywords "colorectal cancer" and "oxidative stress" or "anti-inflammatory" and "IBD." The focus on colorectal cancer is an important contribution.
However, sections 1 and 3 are a superficial description of the topic, and rather a sequence of statements without any detailed or updated insights from underlying research studies. This does not contribute anything useful to the current knowledge. This changes in sections 4.3-4.5, which are fine. Sections 4.1-4.2 need to be improved for clarity and should separate the description of ROS effects in the development of IBD (as a cancer predisposing condition) from those of initiated cancer cells. This would distinguish the review from currently published reviews.
Other comments
1. Abstract: the second and third sentences are very generic and should be removed from the abstract. Instead, two more sentences on specific contents of this review could be added at the end.
2. Fig 2: the legend lacks clarity concerning the distinction between inflammation and chronic inflammation. Also missing are indications on the role of ROS as mutagen for nuclear DNA.
3. Line 97: A sentence or small paragraph is lacking on cellular H2O2 inactivation activities (catalase, glutathione system, etc).
4. Lines 98-102: a short description should be added that ROS are used to modulate signalling pathways in normal cells, however under controlled and locally confined conditions.
5. line 32: ‘leads’ should be ‘lead’ (plural form, as several aspects act all together).
6. line 63: wording: ‘molecules capable of substances oxidation (lipids, proteins, DNA and carbohydrates)’ would be clearer if changed into ‘molecules capable of oxidizing biological substances (lipids, proteins, DNA and carbohydrates)’.
7. Line 66: ‘Prooxidative factors’ is meant to mean the same as pro-oxidant molecules?
8. Line 67: unclear meaning of half-sentence: ‘which process catalysis and the radical degradation of molecules (ionization and UV radiation)’.
9. Line 231: ‘NF-κB p6’ should probably be NF-κB p65.
Round 2
Reviewer 1 Report
The authors carefully addressed the majority of my comments. However, there are still a few issues left with content and grammar, often in the newly changed/added text which have to be addressed.
1. line 28: is it not just "common belief" but there is hard evidence for ROS damaging macromolecules. Also, please remove "the" in front of ROS.
2. Lines 30-31: It is not correct as oxidative modifications ARE changes in bases and can result in sequence changes while strand breaks can also occur directly, without DNA modifications. Moreover, TSG inactivation and oncogene activation occur due to mutations which can result directly from DNA breaks, mistakes in DNA repair etc. in addition to DNA modifications. Please correct and repair.
3. line 54: Grammar is not correct as cells cannot be converted into transformation which is a process. Yu could say "into neoplastically transformed ones/cells...HAS been observed" (conversion is singular). Please correct.
4. line 80: "overview OF/FOR THE roles..."
5. lines 105/6: please add "THE" in front of "glutathion" and "cytoplasm".
6. line 110: better than "positive" say "beneficial"
7. line 234: Rather than "promoting ROS metabolism" which means exacerbating ox stress, you probably mean "counteracting" as this is what antioxidants are doing. Please correct.
8. Similarly, in line 244 it should be "INDUCING" instead of "reducing" antioxidants since otherwise the content ("eliminating ROS" does not make any sense.
9. line 291: Please replace "doses" with "levels" since doses mainly refer to something you externally add and with an exact concentration.
10. line 297: Since there are multiple authors please add "et al.,/and colleagues/co-workers" after "Burwaiss".
11. Line 301: This statement about lower ROS contradicts that about higher ROS in line 290.
12. In the study of ref. 105 hTERT was overexpressed in HeLa cells, thus, these were not normal HeLa cells as your statement suggests while decrease of hTERT expression in these cells increased ROS levels. Please correct, perhaps by saying in line 303 "where by OVEREXPRESSED hTERT..."
13. line 305: remove "the" in front of "telomeres".
14. line 307: Please remove "of" in front of "human"
15. Lines 310-14: Importantly, when mentioning various non-canonical functions of hTERT in relation to oxidative stress, then it is compulsory to describe the fact that it can shuttle to mitochondria where it decreases ROS levels, DNA damage as well as sensitivity to apoptosis (see for example Ahmed et al., 2008 and SInghapol et al., 2013).
16. line 323: Please remove "the" in front of "several"
17. lines 326/7: please replace "which" with "since", otherwise the grammar is wrong. In line 327: replace "the" with "a" in front of "phase".
18. line 355: Please remove "the" in front of "erythrocytes".
19. line 362: OTHER authors" as "another" is only 1, but there are many. Also, order of words "ALSO plays A role..."
20. line 391: "on the BASIS..."
21. line 400 "to A health..."
Reviewer 2 Report
In this revised manuscript, sections 1 and 3 have been slightly improved but the distinction between the role of ROS effects in the development of IBD and of cancer could have been more profound.
Concerning the other 9 comments, the authors have made appropriate changes to respond and this improved the manuscript.
Minor spelling changes:
Line 28: ‘It is commonly belief that’ should be ‘It is commonly believed that’
Line 187: should read ‘oncogenes/tumor suppressor genes’
Line 297: I recommend to write ‘The study of Burwaiss et al. ‘
